# Potential Application of the WST-8-mPMS Assay for Rapid Viable Microorganism Detection

**DOI:** 10.3390/pathogens12020343

**Published:** 2023-02-17

**Authors:** Cheng-Han Chen, Yu-Hsiang Liao, Michael Muljadi, Tsai-Te Lu, Chao-Min Cheng

**Affiliations:** 1Department of Emergency Medicine, Taipei Veterans General Hospital, Taipei 11217, Taiwan; 2Institute of Biomedical Engineering, National Tsing Hua University, Hsinchu 30013, Taiwan; 3School of Medicine, National Yang Ming Chiao Tung University, Taipei 11221, Taiwan

**Keywords:** microorganism detection, colorimetry, point-of-care testing, mPMS, tetrazolium salt, WST-8

## Abstract

To ensure clean drinking water, viable pathogens in water must be rapidly and efficiently screened. The traditional culture or spread-plate process—the conventional standard for bacterial detection—is laborious, time-consuming, and unsuitable for rapid detection. Therefore, we developed a colorimetric assay for rapid microorganism detection using a metabolism-based approach. The reaction between a viable microorganism and the combination of 2-(2-methoxy-4-nitrophenyl)-3-(4-nitrophenyl)-5-(2,4-disulfophenyl)-2H-tetrazolium sodium salt (WST-8) and 1-methoxy-5-methylphenazinium methyl sulfate (mPMS) results in a color change. In combination with a microplate reader, WST-8-mPMS reactivity was leveraged to develop a colorimetric assay for the rapid detection of various bacteria. The detection limit of the WST-8-mPMS assay for both gram-negative and gram-positive bacteria was evaluated. This WST-8-mPMS assay can be used to perform colorimetrical semi-quantitative detection of various bacterial strains in buffers or culture media within 1 h without incubation before the reaction. The easy-to-use, robust, rapid, and sensitive nature of this novel assay demonstrates its potential for practical and medical use for microorganism detection.

## 1. Introduction

Clean water is essential to human beings, and water testing is crucial for providing clean and safe drinking water in areas without sustainable sanitation infrastructure. Unfortunately, most water tests require sophisticated equipment that is inaccessible, complicated, and expensive, and these drawbacks limit their utility in these resource-limited countries [1]. According to the World Health Organization, approximately two billion people do not have access to clean drinking water worldwide [2]. The United Nations International Children’s Emergency Fund has, therefore, urged the development of easy-to-use, rapid, effective, and portable water quality testing devices or kits for bacterial screening and detection in resource-poor areas [3].

The current water bacterial identification techniques, which are laborious and time-consuming, can be categorized into two types: (1) traditional microorganism culture techniques, and (2) biotechnological approaches [4]. Traditional culture methods typically require 24–48 h for incubation, culture, and staining [5,6,7]. Subsequently, the bacterial concentration is measured by counting the number of bacterial colonies in diluted or whole samples [8]. These microorganism culture methods are labor-intensive and time-consuming [5,9]. Biotechnological approaches have been recently introduced to accelerate the identification of microorganisms, such as polymerase chain reaction amplification, matrix-assisted laser desorption ionization time-of-flight spectrometry, and enzyme-linked immunosorbent assays (ELISA) [10,11,12,13,14]. Although these biotechnological approaches provide higher accuracy in specific pathogen identification, they require a period of incubation and should be performed by well-trained personnel with sophisticated equipment [10]. These disadvantages hinder their application in low-socioeconomic and rural areas.

The 3-(4,5-dimethylthiazol-2-yl)-2,5-diphenyltetrazolium bromide-phenazine methosulfate (MTT-PMS) assay is a colorimetric technique that has been traditionally used to evaluate live cell metabolic activity [15]. Previous studies reported that the MTT-PMS assay could provide an instant and semi-quantitative readout of bacterial concentration both in water and human body fluid samples [10,16]. These findings suggest that the MTT-PMS assay can be employed for rapid microorganism screening. However, the formation of insoluble formazan crystals in the MTT-PMS assay may induce apoptosis and cause mechanical damage to cells, thereby potentially interfering with the readout of the MTT-PMS assay [17].

2-(2-methoxy-4-nitrophenyl)-3-(4-nitrophenyl)-5-(2,4-disulfophenyl)-2H-tetrazolium sodium salt (WST-8) is a light-yellow, water-soluble tetrazolium salt. By adding sulfonate groups to benzene rings attached to the tetrazole (WST-8), the formazan derived from the reduction of WST-8 becomes more soluble than the formazan derivatives from MTT in an aqueous solution, suggesting that its use may be more advantageous than that of MTT. Meanwhile, compared with PMS, mPMS (1-methoxy-5-methylphenazinium methyl sulfate) is methoxylated, more stable, and less prone to photodegradation. The combination of the WST-8-mPMS assay, compared with the previously used MTT-PMS assay, may be more effective in detecting bacteria in water. Figure 1 demonstrates the design of the WST-8-mPMS assay and the colorimetric change of the assay with different bacterial concentrations. However, to the best of our knowledge, the efficacy of the WST-8-mPMS assay for microorganism detection has not been evaluated. Therefore, the aim of this study was to evaluate (1) the optimization of the WST-8-mPMS assay and (2) the effectiveness of the WST-8-mPMS assay for microorganism screening.

## 2. Materials and Methods

### 2.1. Reagents and Bacteria

#### 2.1.1. Reagents

(1)WST-8 (Cayman Chemical, Ann Arbor, MI, USA)(2)mPMS (Sigma-Aldrich, St. Louis, MO, USA)(3)Ethylenediaminetetraacetic acid (EDTA): (J.T. Baker Chemical, Phillipsburg, NJ, USA)(4)PBS: phosphate-buffered saline (Sigma-Aldrich, St. Louis, MO, USA)(5)TSB: tryptic soy broth (Sigma-Aldrich, St. Louis, MO, USA)

#### 2.1.2. Bacteria

*Staphylococcus aureus* (TL341), *Escherichia coli* (DH5α), *Klebsiella pneumoniae* (ATCC 23357), and *Streptococcus oralis* (ATCC 6249) were used as the target bacteria in this study. The concentration of bacteria was measured using the NanoDrop Spectrophotometer (Thermo Fisher Scientific, Waltham, MA, USA) with an optical density of 600 nm (OD_600_) = 1, which was considered to contain a concentration of 10^8^ CFU/mL during the experiment.

### 2.2. WST-8-mPMS Assay Optimization

To determine the appropriate concentrations of WST-8-mPMS, 100 μL *E. coli* at a concentration of 10^7^ CFU/mL was added to a transparent 96-well plate before adding 100 μL of different concentrations of WST-8 or the WST-8-mPMS mixture. The mixture was then incubated on the transparent 96-well plate at 37 °C for 2 h. Every 30 min during the incubation, the absorbance of the reagents was measured at 450 nm using the Sunrise Absorbance Microplate Reader (8708, Tecan, Männedorf, Switzerland).

The preparation of each reagent was listed below:(1)Dilute WST-8 reagent in PBS, add 100 μL to a transparent 96-well plate, set the final concentration to 4.5 mM, 2 mM, 0.9 mM, 0.45 mM, and 0.225 mM during the reaction, respectively, and use pure PBS as the blank (0 mM).(2)Dilute WST-8 reagent in PBS, add 100 μL to a transparent 96-well plate, set the final concentration to 4 mM, 2 mM, 0.9 mM, 0.6 mM, 0.45 mM, and 0.2 mM during the reaction, respectively, and use pure PBS as the blank (0 mM).(3)Prepare EDTA reagent in PBS, dilute into 2 × 10^−4^ M, 2 × 10^−5^ M, 2 × 10^−6^ M, and 2 × 10^−7^ M during the reaction, respectively, and use pure PBS as the blank (0 mM).

The following steps were used to optimize the concentration of the WST-8-mPMS assay:(1)Different concentrations of WST-8 reacted with *E. coli*. To evaluate the chronological change in redox reaction between WST-8 and *E. coli*, 100 μL of various concentrations of WST-8 were mixed with 100 μL *E. coli* at a concentration of 10^7^ CFU/mL on the transparent 96-well plate. The mixture was incubated at 37 °C for 2 h, and absorbance was measured at 450 nm every 30 min during the incubation.(2)Different concentrations of mPMS in combination with a single concentration of WST-8 reacted with *E. coli*. To assess the optimal concentrations of mPMS, various concentrations of mPMS were mixed with a single concentration of WST-8. Subsequently, 100 μL WST-8-mPMS mixture and 100 μL 10^7^ CFU/mL *E. coli* were added to the transparent 96-well plate to evaluate the reaction and absorbance. An optimal concentration of mPMS was determined.(3)Different concentrations of WST-8 in combination with a single concentration of mPMS reacted with *E. coli*. To determine the optimal concentration of WST-8, a specified concentration of mPMS, which was selected at the previous step, was mixed with various concentrations of WST-8. 100 μL WST-8-mPMS mixture and 100 μL 10^7^ CFU/mL *E. coli* were added to the transparent 96-well plate to evaluate the reaction and absorbance. The optimal concentration of WST-8 was then determined.(4)Different concentration of EDTA reagent in combination with a single concentration of WST-8-mPMS reacted with *E. coli*. To establish the optimal concentration of EDTA, EDTA was prepared at various concentrations in PBS. Before 100 μL of the WST-8-mPMS mixture was added to each well, 100 μL of *E. coli* at a concentration of 10^7^ CFU/mL was added to a transparent 96-well plate and mixed with various concentrations of EDTA. The mixture was incubated at 37 °C for 2 h, and the absorbance was measured at 450 nm every 30 min during the incubation.

### 2.3. Efficacy of Detection of the WST-8-EDTA-mPMS Mixture for Different Bacterial Species

The common disease-causing pathogens, *E. coli*, *K. pneumoniae*, *S. oralis,* and *S. aureus*, were incubated in TSB at 37 °C to concentrations of 10^8^ CFU/mL by measuring OD_600_ [18,19,20,21]. We then diluted the 10^8^ CFU/mL bacteria with TSB medium to a ten-fold serial dilution, from 10^7^ CFU/mL, 10^6^ CFU/mL, 10^5^ CFU/mL, and 10^4^ CFU/mL to 10^3^ CFU/mL. A pure TSB solution was used as a blank. 100 μL of each concentration of bacteria was mixed with an EDTA solution on a 96-well plate, and then the WST-8-mPMS was added. The plate was then incubated at 37 °C for 2 h. During the incubation, the absorbance at 450 nm was evaluated every 30 min. The limits of detection (LOD) and quantification (LOQ) were calculated using the equation below
$$LOD=Blank\;\left(mean\right)+3\times Blank\;\left(standard\;deviation\right)$$
$$\;\;\;\;\;LOQ=Blank\;\left(mean\right)+10\times Blank\;\left(standard\;deviation\right)$$

### 2.4. Statistical Analysis

All WST-8-mPMS assay results are expressed as CFU/mL. The O.D. results measured by the microplate reader are expressed as the mean ± standard deviation. Comparisons of the mean intensity between different bacterial concentrations were analyzed using the Student’s *t*-test for normally distributed data or the Mann–Whitney U-test for non-normally distributed data. For all statistical results, a *p*-value < 0.05 was considered statistically significant. All statistical analyses were performed using IBM SPSS Statistics for Windows (version 25.0; IBM, Armonk, NY, USA).

## 3. Results

### 3.1. Reduction Ability of WST-8 Concentration on E. coli

Figure 2A shows that the absorbance value at 450 nm increased with increasing concentrations of WST-8 applied to *E. coli* (10^7^ CFU/mL) after one-hour of incubation. The increasing WST-8 concentration was positively correlated with the absorbance at 450 nm, indicating that there was no evident toxicity to *E. coli* was observed (Figure 2A). Figure 2B shows the chronological change in absorbance of different WST-8 concentrations in *E. coli* (10^7^ CFU/mL). After 2 h of incubation with WST-8, the absorbance showed a positive correlation with the concentration of WST-8 (Figure 2B).

### 3.2. Optimization of mPMS Concentration

To investigate the influence of mPMS (an electron-transfer mediator) on the bacterial detection efficacy of WST-8, different concentrations of mPMS were mixed with 0.9 mM WST-8, and 10^7^ CFU/mL *E. coli* were added before incubation. Figure 3A shows the chronological change in absorbance, every half hour during the 2-h incubation, at different concentrations of mPMS (0 mM, 0.2 mM, 0.45 mM, 0.6 mM, 0.9 mM, 2 mM, and 4 mM) mixed with 0.9 mM WST-8. Figure 3B demonstrates that the maximal difference in absorbance was achieved when the concentration of mPMS was at 0.45 mM after 1 h of incubation, compared with the initial absorbance value at 450 nm. When the concentration of mPMS exceeded 0.45 mM, no significant change in the increasing absorbance of the mixed reagents was observed. Meanwhile, when the concentration of mPMS was 0.45 mM, the maximal difference of absorbance was 3.523, which decreased as the concentration of mPMS increased. As a result, an mPMS concentration of 0.45 mM, with the maximal increasing absorbance, was optimal for the WST-8-mPMS assay.

### 3.3. Optimization of the WST-8 Concentration

To ascertain the optimal concentration of WST-8, we treated 10^7^ CFU/mL *E. coli* with 0.45 mM mPMS at different concentrations of WST-8. Figure 4A demonstrates the chronological change in absorbance, every half hour during the 2-h incubation, at different concentrations of WST-8 (0 mM, 0.225 mM, 0.45 mM, 0.9 mM, 2 mM, 4 mM) mixed with 0.45 mM mPMS and 10^7^ CFU/mL *E. coli.*
Figure 4B demonstrates the change in absorbance values at different concentrations of WST-8 mixed with 0.45 mM mPMS and 10^7^ CFU/mL *E. coli* 1 h after 1 h of incubation. Compared to the initial absorbance, the absorbance rapidly increased, whereas the WST-8 concentration that we used was 0.9 mM but subsequently plateaued. A combination of 0.9 mM of WST-8 and 0.45 mM of mPMS was, therefore, adopted as the optimized regimen for bacterial detection.

### 3.4. Effect of EDTA on the Detection of E. coli

Figure 5A demonstrates the chronological change in absorbance after we added different concentrations of EDTA. After 2 h of incubation at 37 °C, the absorbance was higher when the EDTA concentration was 10^−5^ mM. Figure 5B shows the difference in the absorbance values when different concentrations of EDTA were added; at a concentration of 10^−5^ M, a significant increase in absorbance at 450 nm was observed (17%, *p* = 0.005).

### 3.5. Bacterial Detection Ability of WST-8-mPMS

We examined four strains of bacteria: *E. coli* and *K. pneumoniae* (both gram-negative bacteria), *S. aureus* and *S. oralis* (both gram-positive bacteria). The LOD and LOQ were calculated.

Table 1 demonstrates that the LOD of all four species was less than 10^5^ CFU/mL after 2 h of incubation in TSB, and the LOQ was less than 10^5^ CFU/mL for *K. pneumoniae*, *S. aureus*, and *S. oralis* and less than 10^6^ CFU/mL for *E. coli.* Thus, we determined that this reagent combination can be used for the detection of both gram-negative and gram-positive bacteria in TSB. Table 2 shows the bacteria cultured in PBS; the LOQ of all four strains after 1 h of incubation was similar to that of the strains in TSB after 1 h of incubation. Figure 6 shows the different absorbances for bacteria and WST-8-mPMS/EDTA reagents after 2 h of incubation in TSB.

## 4. Discussion

The MTT-PMS assay is a colorimetric approach that has been traditionally used to estimate live cell metabolic activity [15]. Our previous studies demonstrated that the MTT-PMS assay could provide an efficient and semiquantitative readout of the microorganism concentration estimation in water and human body fluids, demonstrating that the MTT-PMS assay may be a potential point-of-care (POC) testing method for rapid live microorganism screening [10,16]. In this study, we first established the optimal design of the WST-8-mPMS assay, which is a modification of the MTT-PMS assay, and then examined the efficacy of WST-8-mPMS for the rapid screening of live bacteria in PBS and TSB. Without incubation before testing, we employed tetrazolium salts, such as MTT or WST-8, to develop a reagent formulation for the rapid estimation of the living microorganism count.

In the MTT-PMS assay, the reduction of MTT catalyzed by the mitochondria is accelerated with the addition of PMS as an electron-transfer mediator [22,23]. Upon catalytic reduction of MTT, quantitation of the generated formazan, which has an intense purple-blue color, can serve as an indicator for the concentration of viable bacteria [15,24]. Unfortunately, the formation of insoluble formazan crystals in the MTT-PMS assay induces apoptosis and causes mechanical damage to the cell envelope [17]. Meanwhile, the insoluble crystal formazan may directly interfere with the results of colorimetric analysis using an ELISA reader, even though the formazan crystals were previously maintained in an alkaline solvent at pH 8, in which insoluble formazan is stably dissolved, as a signal amplifier [16,25]. The disadvantages of the insoluble crystal formazan formed by the reduction of MTT are the main reason why the MTT-PMS assay should be further modified and improved.

The main difference between MTT and WST-8 is that the formazan derived from the reduction of WST-8 remains soluble in an aqueous solution. The use of water-soluble tetrazolium salt has several advantages: (1) unlike MTT, which requires the use of organic or alkaline solvents, the absorbance value can be directly read in an aqueous solution; (2) the generated signal is more efficient than that generated by MTT [26,27]; and (3) formazan crystals that could accumulate inside the cell are not produced, thereby reducing cell toxicity and further lowering the interference of microorganism count estimation [28]. This low-toxicity feature of WST-8 may be ascribed to the generation of water-soluble formazan rather than the cell-damaging insoluble crystals [29]. The increasing WST-8 concentration was positively correlated with the absorbance at 450 nm, indicating that no apparent toxicity to *E. coli* was observed during the redox reaction of WST-8 (Figure 2A).

Moreover, the photosensitive property of PMS, an intermediate electron acceptor of the MTT-PMS assay, may hinder the ease of storage and application of the MTT-PMS assay [30]. Due to the addition of a methoxy group to PMS, mPMS is more stable than PMS. The combination of WST-8 and mPMS, compared with the previously used MTT and PMS, may be more effective in detecting living microorganisms [28]. The addition of mPMS, as an intermediate electron acceptor, can further enhance the reduction of WST-8. However, when the concentration of mPMS exceeded 0.45 mM, no significant increase in absorbance was observed. The absorbance rapidly increased when the WST-8 concentration was 0.9 mM and subsequently plateaued. Therefore, for the optimal WST-mPMS assay, 0.45 mM mPMS was mixed with 0.9 mM WST-8.

As a metal-chelating agent, EDTA can bind to divalent cations such as Ca^2+^ and Mg^2+^ on the cell wall and therefore increase cell permeability [31,32]. We added EDTA to loosen the lipopolysaccharide cell wall of bacteria, thereby increasing cell permeability and shortening the time required for the reagent to pass through the cell wall and react with the enzyme on the cell membrane [33]. Therefore, adding EDTA enhanced the ability of both viable gram-negative and gram-positive bacteria to be detected. Figure 4 shows that the addition of EDTA significantly increased the absorbance after the reaction. We, therefore, added EDTA to the design of the WST-8-mPMS assay.

In this study, we further examined the efficacy of the WST-8-mPMS assay in rapidly screening and estimating different bacteria counts, including gram-positive and gram-negative bacteria. After 1 h of incubation with WST-8-mPMS in TSB medium, the LOQ values for *K. pneumoniae*, *S. aureus,* and *S. oralis* were approximately 10^5^ CFU/mL, and the LOQ for *E. coli* was approximately 10^6^ CFU/mL. However, after 1 h of incubation with WST-8-mPMS in PBS, the LOQ for all species was ~10^6^ CFU/mL. In the MTT-PMS assay, the LOD values for these species were approximately 10^5^ CFU/mL, and the LOD values were about 10^6^ CFU/mL. The LOD and LOQ of the WST-8-mPMS assay were, therefore, not inferior to the MTT-PMS assays and the WST-8-mPMS assay. However, the formazan derived from the reduction of WST-8 becomes more soluble than the formazan derivatives from MTT in an aqueous solution, suggesting that its use may be more applicable in screening microorganisms in water than that of MTT [10,16].

To improve public health and ensure clean water in low-resource settings, the ideal microbial water test device should be portable, self-contained, easily stored, lab-free, electricity-free, low-cost, globally available, and amenable to data communication [1]. *E. coli* can be used as an indicator to monitor the quality of bacterial contamination in water [34,35]. *E. coli*, which is a coliform, can survive for a long time in a contaminated environment and can be used as an indicator of fecal contamination in water [36,37]. The measurement of *E. coli* concentration in water is, therefore, useful and valuable for evaluating water quality [38].

Colorimetric assays are a rapid and timely approach to assessing analytes for healthcare monitoring and water quality testing, ranging from simple pH measurements to pharmaceutical compounds and heavy ion detection [39,40,41,42]. Most colorimetric assays require a laboratory spectrophotometer, such as an ELISA reader, to establish calibration curves through serial absorbance measurements. The analyte concentration can then be calculated quantitatively by comparing the absorbance at specific wavelengths with the designated calibration curve [43]. However, these spectrophotometric devices, which are expensive and require a laboratory, are not suitable for POC testing. To ensure that colorimetric assays are portable and readily available for POC testing employed in resource-limited regions, colorimetric assays are usually integrated into a paper-based analytical device, thus providing a qualitative result. The MTT assay, as an example, has been integrated into paper-based devices to estimate sperm motility [44]. Furthermore, with the integration of smartphone and RGB analysis applications, such as ImageJ, paper-based colorimetric assays can further provide a semi-quantitative or quantitative result [45,46,47].

Several other rapid water quality testing devices have been reported. These approaches include the following: a capacitive matrix biosensor capable of measuring differentiated *E. coli* concentrations based on matrix topology variability [48]; immunodetection via a microfluidic biosensor linking gold nanoparticles [47]; and a paper-based biosensor with a handheld heating device for detecting nucleic acids combined with loop-mediated isothermal amplification [49]. The reported LOD for each approach varies from 50 CFU/mL to 10^5^ CFU/mL [47,48,49,50]. Table 3 shows the comparison of these methods to our WST-8-mPMS assay. Although the aforementioned techniques have more sensitive detection limits, they were designed for specific pathogens, which limits their utility for screening a wide range of gram-negative and gram-positive bacteria. Our study demonstrated that the WST-8-mPMS assay has acceptable LODs and LOQs for gram-positive and gram-negative pathogens. The WST-8-mPMS assay detects “viable” microorganisms because of its metabolic reaction with live pathogens, and this metabolic activity approach can also be employed in detecting viable but nonculturable (VBNC) microorganisms, which are also critical pathogens in food or water contamination [15,24,51]. Furthermore, similar to the MTT-PMS assay, the WST-8-mPMS assay has the potential to generate a semiquantitative colorimetric result on paper-based POC devices, thus allowing the rapid screening of live microorganisms in water.

Thus, we designed a novel, quick, and viable microorganism detection method with the WST-8-mPMS assay; however, this assay has a few limitations. First, the study was performed in a buffer system; therefore, the utility of the WST-8-mPMS assay in water or even body fluids should be further evaluated. Furthermore, previous studies have reported that the presence of heme might interfere with the spectral absorption and the redox reaction of the tetrazolium salt, causing either false-positive or false-negative results [52,53,54]. The interference of heme or hemoglobin in the blood hinders the viability of the MTT-PMS assay to be utilized as a rapid bacterial detection tool for blood samples; however, the interaction between the WST-8-mPMS assay and heme remains unknown. Additional studies are required to investigate heme interference and broaden the utility of the WST-8-mPMS assay, especially for use with samples containing blood. Furthermore, the LOD of the WST-8-mPMS assay was approximately ~10^6^ CFU/mL, a value that could be improved by further optimizing the elements of the WST-8-mPMS assay. Finally, as a screening assay, the WST-8-mPMS assay is unable to identify specific microorganisms; therefore, it is only useful to quantify the total amount of microorganisms. In contrast, it cannot quantify the concentrations of each specific pathogen.

## 5. Conclusions

This is the first study to report that the WST-8-mPMS assay could be a potential rapid detection method for live microorganism screening, which might be crucial for testing water contamination. Considering the stability, storage time, preparation, measurement convenience, and lower detection limit of the WST-8-mPMS assay, this assay may be preferable to the traditional MTT assay and may be more useful for microorganism screening. The characteristics of the WST-8-mPMS assay, similar to the MTT-PMS assay, can be further integrated into paper-based POC testing systems as a useful “triage” tool for screening live pathogens prior to routine culture techniques or even detecting the presence of VBNC microorganisms. Further research is required to investigate the broader applicability of WST-8-mPMS assay-based microorganism detection.

## Figures and Tables

**Figure 1 pathogens-12-00343-f001:**
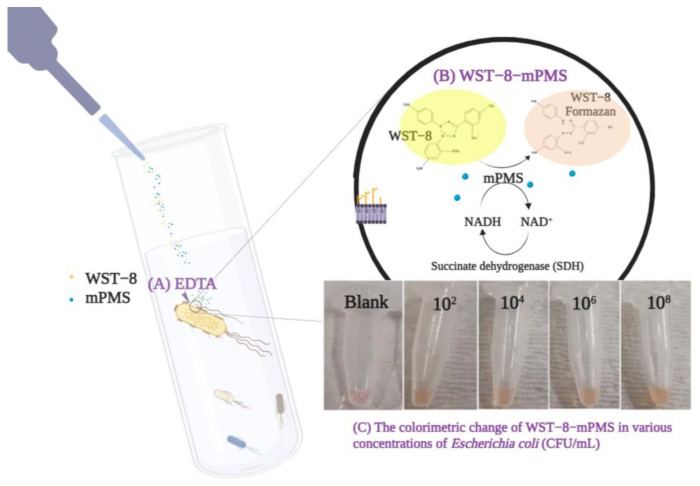
Design of the WST-8-mPMS assay. (**A**) Ethylenediaminetetraacetic acid (EDTA) is added to the target solution to enhance the permeability of the bacterial cell walls; (**B**) the WST-8-mPMS mixture is added to the solution. As the WST-8-mPMS reacts with cellular succinate dehydrogenase, the light-colored tetrazolium salt is reduced to a soluble orange formazan form. (**C**) The colorimetric change of WST-8-mPMS at various concentrations of *Escherichia coli* (colony-forming units [CFU]/mL). The figure was created using BioRender.com (accessed on 14 February 2023).

**Figure 2 pathogens-12-00343-f002:**
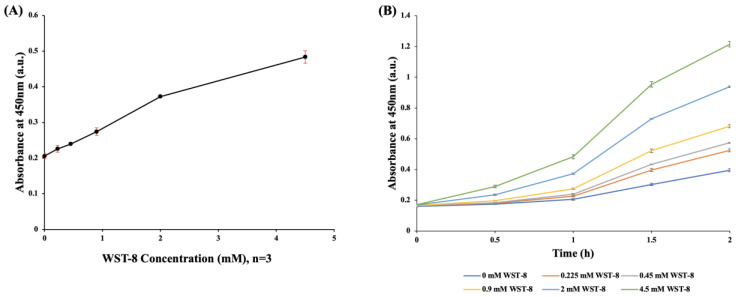
Reduction ability of WST-8 concentration on *E. coli.* (**A**) Changes in absorbance at different concentrations of WST-8 after 1 h of incubation with *E. coli* (10^7^ CFU/mL). Higher WST-8 concentrations (0 mM, 0.225 mM, 0.45 mM, 0.9 mM, 2 mM, and 4.5 mM) showed increased absorbance at 450 nm (0.2053, 0.2257, 0.2393, 0.274, 0.3723, and 0.4833, respectively); (**B**) Chronological changes in absorbance at different concentrations of WST-8 incubated with *E. coli* (10^7^ CFU/mL). Higher WST-8 concentrations (0 mM, 0.225 mM, 0.45 mM, 0.9 mM, 2 mM, and 4.5 mM) showed increased absorbance at 450 nm after 2 h of incubation (0.236, 0.362, 0.410, 0.516, 0.768, and 1.045, respectively).

**Figure 3 pathogens-12-00343-f003:**
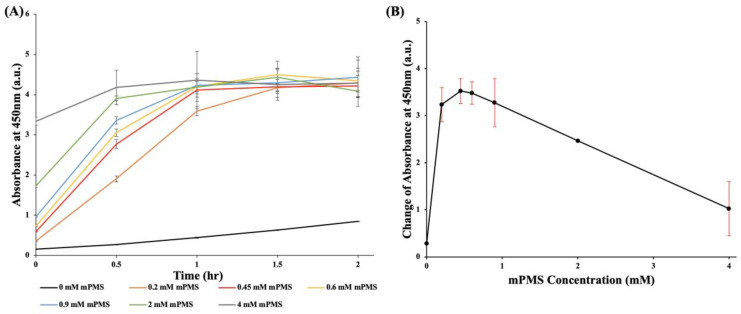
Optimization of mPMS concentration. (**A**) The chronological change in absorbance at different concentrations of mPMS (0 mM, 0.2 mM, 0.45 mM, 0.6 mM, 0.9 mM, 2 mM, and 4 mM) with 0.9 mM WST-8 and *E. coli* (10^7^ CFU/mL). After 1 h of incubation, the absorbances were 0.441, 3.591, 4.114, 4.203, 4.232, 4.190, and 4.363. After 2 h of incubation, the absorbances were 0.8447, 4.2983, 4.2147, 4.348, 4.4353, 4.088, and 4.2857. (**B**) Change of absorbance at different concentrations of mPMS (0 mM, 0.2 mM, 0.45 mM, 0.6 mM, 0.9 mM, 2 mM, and 4 mM) with 0.9 mM WST-8 and *E. coli* (10^7^ CFU/mL). After 1 h of incubation, the absorbances increased by 0.2827, 3.234, 3.5227, 3.4793, 3.276, 2.465, and 1.0233, respectively.

**Figure 4 pathogens-12-00343-f004:**
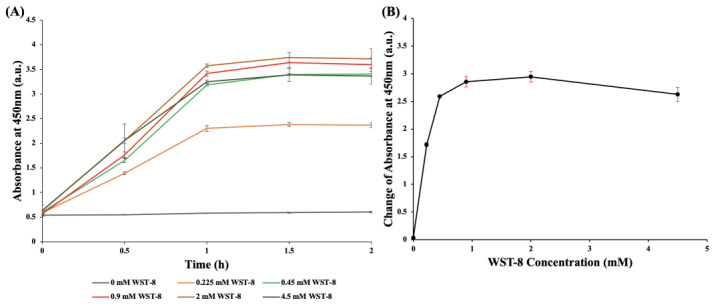
Optimization of the WST-8 concentration. (**A**) The chronological change in absorbance at different concentrations of WST-8 with 0.45 mM mPMS and *E. coli* (10^7^ CFU/mL). After 1 h of incubation, the absorbances were 0.579, 2.303, 3.1873, 3.4153, 3.576, and 3.2513. After 2 h of incubation, the absorbances were 0.6023, 2.366, 3.4083, 3.5953, 3.7167, and 3.3667. (**B**) The change in absorbance of different concentrations of WST-8 (0 mM, 0.225 mM, 0.45 mM, 0.9 mM, 2 mM, 4.5 mM) with 0.45 mM mPMS and *E. coli* (10^7^ CFU/mL). After 1 h of incubation, the change of the absorbance, compared to the initial value, were 0.0353, 1.718, 2.5903, 2.8533, 2.945, and 2.6257.

**Figure 5 pathogens-12-00343-f005:**
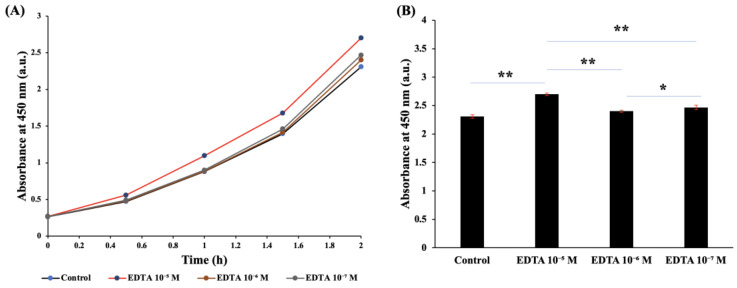
The effect of EDTA on the detection of *E. coli.* (**A**) Chronological changes in absorbance at different concentrations of EDTA with 0.45 mM mPMS and 0.9 mM WST-8. After 2 h of incubation with 10^7^ CFU/mL *E. coli*, the absorbances were 2.307, 2.700, 2.401, and 2.4675. (**B**) The absorbance at different concentrations of EDTA with 0.45 mM mPMS and 0.9 mM WST-8. After 2 h of incubation with *E. coli* (10^7^ CFU/mL), the absorbances were 2.307, 2.700, 2.401, and 2.4675. * *p* < 0.05, ** *p* < 0.01.

**Figure 6 pathogens-12-00343-f006:**
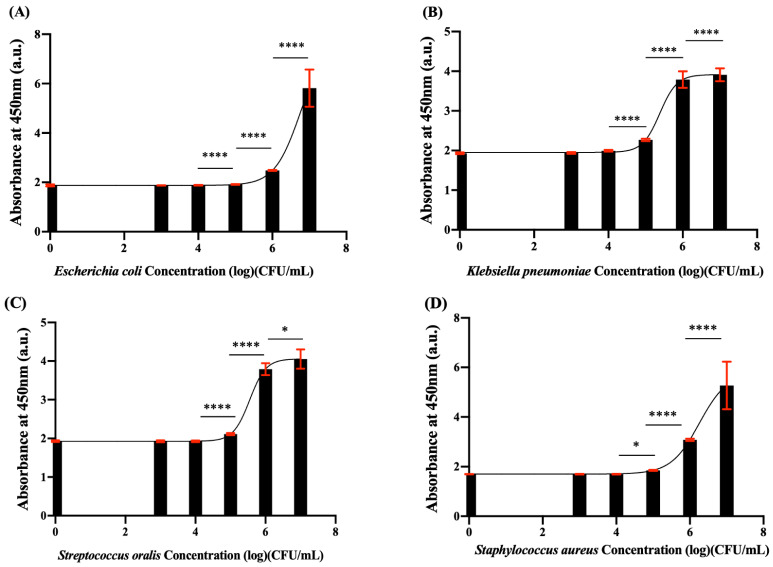
Bacterial detection ability of WST-8-mPMS in TSB. (**A**) The absorbance at different concentrations of *E. coli* incubated with WST-9-mPMS/EDTA for 2 h; (**B**) The absorbance at different concentrations of *K. pneumoniae* incubated with WST-8-mPMS/EDTA for 2 h; (**C**) The absorbance at different concentrations of *S. oralis* incubated with WST-8-mPMS/EDTA for 2 h; (**D**) The absorbance at different concentrations of *S. aureus* incubated with WST-8-mPMS/EDTA for 2 h. * *p* < 0.05, **** *p* < 0.0001.

**Table 1 pathogens-12-00343-t001:** LOD and LOQ of the WST-8-mPMS assay versus various bacteria in tryptic soy broth (TSB) medium.

	0.5 h LOD	0.5 h LOQ	1 h LOD	1 h LOQ	2 h LOD	2 h LOQ
*E. coli*	7.27 × 10^5^	1.25 × 10^6^	5.37 × 10^5^	1.04 × 10^6^	4.01 × 10^4^	6.96 × 10^5^
*K. pneumoniae*	2.00 × 10^5^	6.03 × 10^5^	8.39 × 10^4^	2.73 × 10^5^	4.12 × 10^4^	8.19 × 10^4^
*S. aureus*	8.82 × 10^5^	1.12 × 10^6^	2.13 × 10^5^	5.50 × 10^5^	2.20 × 10^4^	6.49 × 10^4^
*S. oralis*	7.47 × 10^5^	9.53 × 10^5^	2.03 × 10^5^	4.67 × 10^5^	5.75 × 10^4^	1.14 × 10^4^

*E. coli*: Escherichia coli; *K. pneumoniae*: Klebsiella pneumoniae; *S. aureus*: Staphylococcus aureus; *S. oralis*: Streptococcus oralis.

**Table 2 pathogens-12-00343-t002:** LOD and LOQ of the WST-8-mPMS assay versus various bacteria in phosphate-buffered saline (PBS).

	0.5 h LOD	0.5 h LOQ	1 h LOD	1 h LOQ
*E. coli*	5.55 × 10^5^	7.34 × 10^6^	6.57 × 10^5^	8.17 × 10^5^
*K. pneumoniae*	2.23 × 10^5^	4.13 × 10^5^	2.08 × 10^4^	3.61 × 10^5^
*S. aureus*	8.03 × 10^5^	1.00 × 10^6^	7.17 × 10^5^	9.01 × 10^5^
*S. oralis*	6.18 × 10^5^	7.58 × 10^5^	6.14 × 10^5^	7.49 × 10^5^

**Table 3 pathogens-12-00343-t003:** Comparison of different bacterial screening tests.

Method	Time	LOD	Equipment and Technique
Microwave matrix sensor	A few hours	10^3^ CFU/mL for *E. coli*	Microwave sensor and semiconductive matrix sensor [48]
Gold nanoparticle colorimetric biosensor	A few hours	10^3^ CFU/mL for *E. coli* O157	Colorimetric gold nanoparticle (GNP) biosensors and optical biosensors [47]
Integrated paper-based nucleic acid testing	1 h	10–10^3^ CFU/mL for *E. coli* and *Streptococcus pneumoniae*	Loop-mediated isothermal amplification and gold nanoparticle conjugation [49]
WST-8-mPMS assay	1 h	10^5^–10^6^ CFU/mL for live microorganisms	Colorimetric analysis by the tetrazolium reduction and spectral sensors

## Data Availability

The datasets generated during the current study and the developed code are available from the corresponding author on reasonable request.

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
