# Peer review of "Potential Application of the WST-8-mPMS Assay for Rapid Viable Microorganism Detection"

_pathogens, 2023, doi:10.3390/pathogens12020343_

Round 1

Reviewer 1 Report

The manuscript pathogens-2197751 “Potential Application of the WST-8-mPMS Assay for Rapid Viable Microorganism Detection”, describes the development of a colorimetric assay for rapid microorganism detection, which could be surely useful for bacterial screening and detection especially in resource-limited countries.

The study is interesting and the applied approach seems adequate; however, I would suggest a revision especially in “materials and methods” and in “results” sections.

INTRODUCTION

Lines 26 – 41: The first paragraph could be reorganized and improved. You could start the paragraph talking about “infectious diseases” (e.g. from line 29 to line 38 “in resource-limited or (…) ensured”), then you can introduce the concept of “water testing” (e.g. from line 26 to line 29 “clean water….countries”), finally you can conclude talking about the need underlined by UNICEF (from line 39 to line 41).

Lines 27 – 29: The term “which” appears two times, please revise the sentence.

Lines 65 - 68: It should be emphasized that the application of WST-8 could be more advantageous than the application of MTT. Maybe you can add at the end of the phrase “, so its application could be more advantageous with respect to the application of MTT” .

Lines 75 – 76: “to evaluate the optimal design” sounds strange, maybe you can change as “to optimize/set up the WST-8-mPMS assay and 2) to assess the …”

Figure 1: The caption of figure 1 should be improved, it seems a little repetitive. In the part A) you can say that EDTA solution and the WST-8-mPMS mixture are added (explaining the role of EDTA) (do not talk about succinate dehydrogenase). In part B) you can describe the reaction of WST-8-mPMS with the succinate dehydrogenase and the reduction of the WST-8. Please revise also the figure: Escherichia coli should be in italics.

MATERIALS AND METHODS

Lines 93 – 96: Put the acronyms after the words. E.g. ethylenediaminetetracetic acid – EDTA (…).

Lines 95 – 96: Add “St. Louis, MO, USA” after “Sigma-Aldrich” in.

Line 101: Please, change “number” with “concentration” and revise the sentence. “And regarded as” sounds strange and the sentence is unclear. Perhaps you meant that a bacterial suspension with an OD at 600 nm equal to 1 was considered to contain a concentration of bacteria of 10^8 CFU/mL?

Line 105/106: Please revise the titles removing “to determine” . Maybe an appropriate title could be “WST-8-mPMS optimization/set up”. You can put together paragraph 2.2.1 with paragraph 2.2.2.

Lines 107, 110, 115, 116, 117, 118 etc.: Please change the sentences removing the term “we” (use passive form).

Lines 109, 110 etc.: Was the plate that you used in the assay an ELISA plate or was it a “standard” transparent 96-well plate? Please check, talking about ELISA plate could be confusing if this assay is not an ELISA assay (I think that this assay is not an ELISA assay).

Paragraph 2.2.1/ 2.2.2: Please revise the paragraph. I can not understand the steps that were used to perform the assay. In the first part you stated that E.coli was added and then you added WST-8 alone or a mixture of WST-8 and mPMS. After you stated that you added E.coli to the mixed reagents (i.e. the reagents were added before the addition of E.coli). Was E. coli added before or after the reagents? Maybe it could be better to reorganize the paragraph. Below my suggestion.

Begin the paragraph by describing how the solutions of WST-8, mPMS and EDTA were prepared (i.e. in PBS, as reported in lines 122- 123); describe also the concentrations of each reagent that were tested. [Check the concentrations of the WST-8: in figure 2B and 4B the reported concentrations are 0.2, 0.45, 0.6, 0.9, 2, 4 mM, while in “materials and methods” sections the values are 0.4, 0.9, 1.2, 1.8, 4, 9 mM (lines 114 – 115); add the used concentrations of mPMS that are lacking]. Next, describe the 4 experiments that were performed. Please describe them using the order that you used to describe the results: 1) different concentrations of WST-8 alone, 2) different concentrations of mPMS in combination with a single concentration of WST-8, 3) different concentrations of WST-8 in combination with a single concentration of mPMS, 4) different concentrations of EDTA in combination with a single concentration of WST-8 and mPMS.

Lines 129 – 130: Were the pathogens incubated in TSB? What were the incubation conditions (temperature, medium)?  Please specify.

Lines 130 – 131: “the optimal concentrations….PBS” remove the sentence, it is repetitive. 

Lines 139: Please specify the meaning of the terms included in the equation (i.e. bottom, top, ec50 etc.). What is the relationship between this equation and LOD/LOQ? The LOD/LOQ were calculated with the equations reported in lines 226 – 227, maybe these equations should be moved to this section (i.e. “materials and methods” section).

Lines 147 – 150: “this section may be divided…” remove these sentences.

RESULTS

Figures 2 – 6 and figure captions

Maybe it should be better to remove the title of each graph; the details reported in the titles should be reported in figure captions and/or in x-axis, y-axis and legend. If you want to report some additional details (now reported in titles) in figures, you can use the following suggestions:

·         In figure 2B “ WST-8” can be added in the legend to clarify that the concentrations refer to WST-1 (as you did in figure 5A for EDTA).

·         In figure 3B “mPMS” can be added in the legend to clarify that the concentrations refer to mPMS (as you did in figure 5A for EDTA).

·         In figure 4B “ WST-8” can be added in the legend to clarify that the concentrations refer to WST-1 (as you did in figure 5A for EDTA).

·         In figure 6A – D the name of the bacterium can be reported in the x-axis (e.g. “ E. coli concentration [log(CFU/ml)]”

Check if the “@” in the y-axis is correct. Maybe you can write “at 450 nm” or write “450 nm” as a subscript.

Use the symbol “h” for the measure unit of time (now it is reported as “hours/HOURS/hr”).

Lines 166 – 167: Please add the incubation time at which the reported absorbance values were measured (1.5 h?) or remove the whole sentence.

Figure 3A and 3B: Should the values shown in figure 3B (after 1h of incubation) be the same or similar to those shown in figure 3A? why are they so different? Different values are also shown in the caption (line 183 and 186). Please check and explain.

Line 182 and line 185: The concentration of 0.6 mM is not reported in the figure caption but it appears in the figures 3A and 3B.

Lines 163, 166, 182, 185, 202: Remove “from” in the parenthesis.

RESULTS (text)

Line 178: “or 0.45 mM, 0.6 mM, 0.9 mM” Why the concentrations of 2 and 4 mM are not reported in the parenthesis?

Line 194 – 195: Please revise the sentence “the absorbance rapidly increased when the WST-8 concentration was 0.9 mM”. The use of “when” seems to be inappropriate.

Line 223: Please do not abbreviate S. aureus and S. oralis as “S.a.” and “S.o.” but write them as “S. aureus” and “S. oralis” (revise table 1 and table 2).

Lines 233 – 234: Add the incubation time “the LOQ of all four strains AFTER 1 H incubation…”.

Lines 235 – 236: Add “in TBS”. Add “in TBS” also in the caption of figure 6.

Lines 249 – 250: “Detection” is repeated two times, revise.

Table 2: Why did you not test LOD and LOQ after 2h?

DISCUSSION

Lines 258 – 278: The first part of the discussion should be reorganized. Here some suggestions. Start the section with the phrase “the MTT-PMS assay… activity” (lines 260 – 262), then put the phrase “ Our previous study….screening” (lines 264 – 267), then continue with “ In the MTT-PMS assay……improved” (lines 268 – 278), and finally report the aim of the study “ In this study…. TSB” (lines 258 – 260) and “Without incubation…. Count” (lines 262 – 264). In the last phrase “without incubation…. Count” (lines 262- 264) remove “MTT or”.

Lines 359 – 371: Maybe, among the limitations, you should mention that the proposed assay is not able to identify specific microorganisms so it is only useful to quantify the total amount of microorganisms. On the contrary, it cannot quantify the concentrations of each specific pathogen.

Reviewer 2 Report

Brief summary

In this paper the authors describe the results of implementation of a novel method to detect viable bacteria based on WST-8-mPMS assay.

Broad comments

In my opinion the topic is interesting especially if related to the necessity to have rapid results relating to microbial contamination of the water. However, the authors presented experimental data obtained by lab scale tests, so there are no data (as underlined in the limit of the study part) regarding the potentiality of the assay in its application on water in the environment. In the introduction section the authors have to not emphasized a lot this aspect but addressed to the novelty of this assay that could be used for viable bacterial detection.

Moreover the tested bacteria are not typical of water pollution, except for E.coli. In methods section the authors must justified the choice of microbial target used for the tests. Why specific pathogenic bacteria are not tested (e.g. Salmonella)?

Reviewer 3 Report

In this article, Chen et al. developed a WST-8-mPMS assay for bacterial detection in water based on the traditional MTT-PMS assay. In addition, Chen investigated the effects of reduction ability, Wst-8 concentration, mPMS concentration and EDTA, and assessed the bacteria detection ability of WST-8-mPMS. This study has implications for the application of improving public drinking water.

1.     The experimental design appears to be incomplete and lacks the comparative data on the bacterial detection ability of WST-8-mPMS and MTT-PMS, in order to confirm the superiority of the WST-8-mPMS assay.

2. In the introduction section, the author mentioned that the insoluble crystals formed in the traditional MTT-PMS assay may cause cell damage, but the author do not seem to characterize the effect of the WST-8-mPMS assay on cell damage in this study.

3. we found multiple fonts in the figures, please make them consistent throughout, e.g., Arial.

       4. Please check the grammatical errors throughout the article.

Round 2

Reviewer 1 Report

The manuscript has been improved so it is now suitable for publication.

Reviewer 3 Report

The revised manuscript can be considered for publication.